# GRO-RAG: Gradient-aware Re-rank Optimization for Multi-source Retrieval-Augmented Generation

**Siyuan Chen**[1,*]    **Ding Hang**[2,*]    **Xiaoyu Kang**[3]    **Jiechao Gao**[4,†]

[1]University of Bristol
[2]Shanghai Jiaotong University
[3]Institute of Information Engineering, Chinese Academy of Sciences
[4]Stanford University

## Abstract

Retrieval-Augmented Generation (RAG) systems often rely on information retrieved from heterogeneous sources to support generation tasks. However, existing approaches typically either aggregate all sources uniformly or statically select a single source, neglecting semantic complementarity. Moreover, they commonly employ re-ranking models to obtain Top-k documents, without accounting for actual contribution to generation objective. In this paper, we propose GRO-RAG, a training-free, gradient-aware re-ranking framework for multi-source RAG. Our method performs Top-k document selection by reading gradients from the language model, estimating each document's contribution to the generation loss through a single backward pass. This enables re-ranking not by heuristic relevance, but by direct feedback from LLM's generation objective. At the source level, we incorporate inter-source redundancy and query relevance to select source combination prior to re-ranking. Theoretically, we prove that this gradient-based Top-k selection approximates the optimal subset minimizing the generation loss, and aligns with minimizing the leave-one-out loss upper bound. Experiments across multi-source QA and open-domain generation tasks demonstrate consistent improvements in generation quality, highlighting the importance of generation-aware retrieval selection in multi-source RAG.

## 1 Introduction

Retrieval-Augmented Generation (RAG) (Lewis et al., 2020) has emerged as a powerful paradigm for enhancing large language models (LLMs) by grounding their outputs in external knowledge. A typical RAG pipeline (Chen et al., 2017; Das et al., 2019) first retrieves a set of supporting documents from a corpus, and then conditions the generation process on both the query and the retrieved context. In practice, especially in open-domain and multi-hop settings, information is often distributed across multiple heterogeneous sources such as encyclopedias, web documents (Komeili et al., 2022; Dinan et al., 2019), or community forums. This gives rise to the challenge of multi-source retrieval, where the system must identify not only relevant documents, but also determine which sources to trust, combine, or ignore (Yan et al., 2024; Yao et al., 2023; Wang et al., 2024; Zhao et al., 2024a).

By integrating multiple retrieval sources, Multi-Source RAG mitigates the capability ceiling limitation inherent in Single-Source RAG. Recent studies (Yan et al., 2024; Yao et al., 2023; Wang et al., 2024; Zhao et al., 2024a) demonstrate that leveraging multiple retrieval sources dynamically and controllably can improve retrieval accuracy, thereby enabling the generation of more comprehensive and high-quality knowledge-grounded responses. For instance, ReAct (Yao et al., 2023) introduces an iterative multi-source retrieval approach, whereas UniMS-RAG (Wang et al., 2024) uniquely integrates source selection, retrieval, and generation into a unified model enhanced with action and evaluation tokens,

---

[*]Equal contribution.
[†]Corresponding author: `jiechao@stanford.edu`

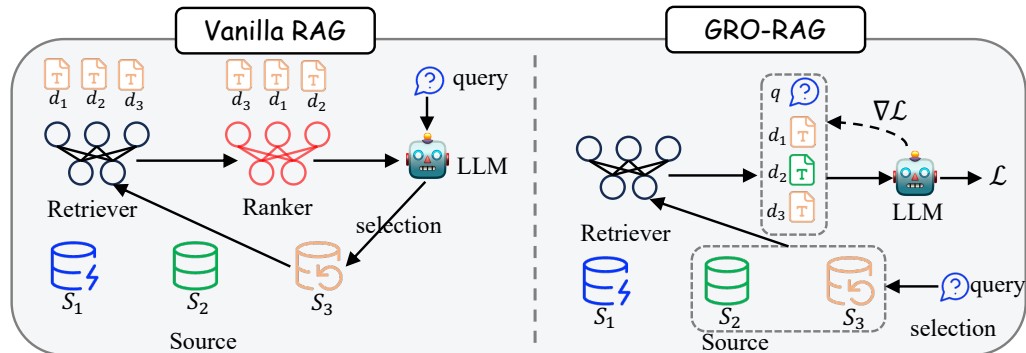

Figure 1: Comparison between vanilla RAG and our proposed GRO-RAG. Vanilla RAG pipelines retrieve documents from certain source and rank them based on query-document relevance. GRO-RAG selects a subset of sources that jointly balance query relevance and semantic diversity. And it re-ranks retrieved documents using a single backward pass over the generation loss, estimating each document's contribution via the alignment between its representation and the loss gradient.

enabling the language model to dynamically invoke and filter sources based on real-time demands. Additionally, PrefRAG (Zhao et al., 2024a) advances RAG further by employing a preference-driven adaptive retrieval mechanism coupled with self-reflection, supporting in-depth and controlled exploration across diverse retrieval sources. Despite these advances, most existing RAGs treat source-level retrieval in a simplistic manner—either aggregating all sources uniformly or statically selecting a single source—thereby neglecting the semantic diversity and redundancy inherent across sources. Furthermore, even after document retrieval, many systems (e.g., BM25 (Robertson et al., 2009)) rely on re-ranking models that score documents solely based on retrieval-level signals such as query-document similarity, without considering their actual utility to the downstream generation objective. This creates a mismatch between what is retrieved and what is ultimately needed for high-quality generation.

To address these limitations, we propose GRO-RAG, a training-free optimization framework tailored for Multi-source Retrieval-Augmented Generation (MS-RAG) (Yan et al., 2024; Yao et al., 2023; Wang et al., 2024; Zhao et al., 2024a). GRO-RAG introduces a principled mechanism to dynamically select both *"which source combinations to retrieve from"* and *"which documents to use as context"*. Firstly, we select source combinations by optimizing a relevance-redundancy tradeoff that jointly considers query relevance and inter-source semantic overlap, rather than uniformly aggregating or statically selecting sources. This allows GRO-RAG to identify a diverse yet relevant set of sources that provide complementary information, improving both recall and retrieval quality at the source level. Subsequently, we introduce a gradient-aware document re-ranking strategy that directly leverages the feedback from large language model. Specifically, after retrieving candidate documents from the selected sources, we perform a forward pass to compute the generation loss and then a single backward pass to obtain the gradient of this loss with respect to the input representations. For each document, we compute an importance score as the inner product between its hidden representation and the loss gradient. This score reflects how much the document contributes to reducing the generation loss. Selecting the Top-k documents based on these scores enables a posterior-aware re-ranking process that is tightly aligned with the final generation objective, rather than heuristic query-document similarity. GRO-RAG is fully training-free and compatible with frozen language models. It introduces no additional model parameters or training phases, and requires only one forward-backward pass per query. Theoretically, we show that the gradient-based Top-k selection approximates the solution to an underlying utility maximization problem and aligns with minimizing a leave-one-out loss upper bound.

Our main contributions are summarized as follows:

- We introduce a novel, training-free approach that estimates each document's contribution to the generation loss using a single backward pass, enabling posterior-aware Top-k selection based on model feedback.

- We formulate source selection as a relevance-redundancy tradeoff over source subsets, enabling dynamic source combination and improving retrieval diversity and complementarity.

- We theoretically show that our gradient-based selection approximates an underlying utility maximization objective and aligns with minimizing a leave-one-out loss upper bound. Extensive experiments on multi-source QA and open-domain generation benchmarks demonstrate consistent improvements over strong retrieval and re-ranking baselines.

## 2 RELATED WORKS

### 2.1 RETRIEVAL-AUGMENTED GENERATION

Retrieval-augmented generation (RAG) is increasingly recognized as an effective method to mitigate several limitations of large language models (LLMs) (Wu et al., 2025; Tang et al., 2023), notably hallucinations (Shuster et al., 2021), factuality issues (Wang et al., 2023), and the lack of long-term memory (Xu et al., 2022). Following the established "Retriever-and-Reader" paradigm (Chen et al., 2017; Das et al., 2019), RAG first employs an external retriever to select relevant textual information from knowledge sources(e.g., Wikipedia). These retrieved passages subsequently serve as external context for a reader/generator, enabling the model to generate knowledge grounded response (Lewis et al., 2020). Initial retrieval methods (e.g., BM25 (Robertson et al., 2009)) used sparse retriever for relevance scoring, but often fail to capture deeper semantic information (Guo et al., 2022). To overcome this limitation, language-model-based dense retrieval approaches have been developed, encoding documents and queries into dense vectors to effectively represent the semantic feature of text content (Karpukhin et al., 2020; Li et al., 2023; Bruch et al., 2023). Recently, researchers have explored leveraging LLMs as retrievers (Wang et al., 2024; Asai et al., 2023; Jiang et al., 2023; Yu et al., 2024; Zhao et al., 2024a). For instance, Self-RAG introduces reflection tokens, enabling the model to dynamically retrieve supporting passages and self-correct its outputs (Asai et al., 2023).

### 2.2 MULTI-SOURCE RAG

Based on the retrieval sources, recent advances in retrieval-augmented generation (RAG) can be categorized into Single-Source RAG (SS-RAG) (Asai et al., 2023; Jiang et al., 2023; Yu et al., 2024; Cui et al., 2025) and Multi-Source RAG (MS-RAG). SS-RAG methods inherently limit RAG system performance due to reliance on a single knowledge source. MS-RAG addresses this limitation by integrating heterogeneous knowledge sources, including specialized databases, structured archives, and the open web (Yan et al., 2024). Common MS-RAG implementations perform sequential or parallel retrieval across multiple indices. For example, CRAG treats the web as a fallback source (Yan et al., 2024), while ReAct coordinates retrieval and reasoning within an agent-based framework (Yao et al., 2023). However, naively concatenating evidence from diverse sources can enlarge context windows and introduce noise or conflicting information, thus necessitating *adaptive source selection* (Wang et al., 2024; Zhao et al., 2024a). UniMS-RAG (Wang et al., 2024) addresses these issues by unifying source selection, retrieval, and generation into a single sequence-to-sequence model. It introduces specific action and evaluation tokens, allowing the LLM to dynamically invoke and filter sources as needed. PrefRAG (Zhao et al., 2024a) further advances MS-RAG with a preference-driven adaptive retrieval approach that employs self-reflection, enabling more in-depth and controllable exploration across multiple retrieval sources.

## 3 METHOD

### 3.1 TASK DEFINITION AND NOTATION

Let $\mathcal{Q}$ be the space of user queries and $\mathcal{A}^*$ the space of target answers. We are given a query $q \in \mathcal{Q}$ and seek to generate an answer $a^*$ with the assistance of external evidence. Evidence is organized in a set of *heterogeneous sources* $\mathcal{S} = \{s_1, \ldots, s_{|\mathcal{S}|}\}$. Each source $s \in \mathcal{S}$ exposes a (possibly dynamic) document collection $\mathcal{D}_s = \{d_{s,1}, \ldots, d_{s,|\mathcal{D}_s|}\}$, where a document $d$ is a sequence of tokens from the vocabulary $\mathcal{V}$. Our goal is to choose both (i) a subset of sources $\mathcal{A}^\star \subseteq \mathcal{S}$ and (ii) a size-$k$ document set $D^\star(q) \subseteq \bigcup_{s \in \mathcal{A}^\star} \mathcal{D}_s$ that minimizes the generation loss. A frozen Transformer (Vaswani et al., 2017) language model $\mathcal{M}_\theta$ receives the query $q$ together with a small context set $D^*$ and produces a

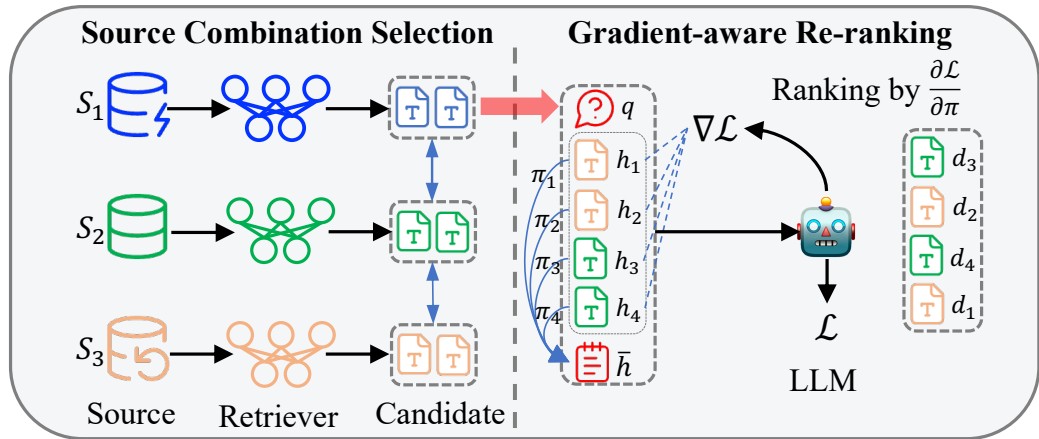

Figure 2: Model architecture of GRO-RAG

best answer $a^*$. Because enumerating all subsets is infeasible, the following two subsections describe a tractable, training-free procedure—first selecting sources, then ranking documents with gradient feedback—that approximates the optimum of generation objective.

## 3.2 SOURCE COMBINATION SELECTION

Suppose a geography query asks *"Which European river flows through both Vienna and Bratislava?"*. A news feed, Wikipedia, and a travel blog may each contain statements answering the question. If we indiscriminately merge all three sources the model will read many redundant sentences about the Danube, wasting context budget. Conversely, if we keep only Wikipedia, we indeed cover the Danube, but we throw away the travel-blog anecdote that might help the model answer follow-up questions (e.g. travel time, boat service). Our goal is therefore to retain sources that bring new information while discarding those that merely repeat what has already been covered.

Selecting *which source combinations to retrieve from* is crucial in multi-source RAG (Wang et al., 2024; Zhao et al., 2024a): querying too many sources balloons the candidate pool with near-duplicates, whereas restricting to a single source squanders complementary evidence that may be indispensable for multi-hop reasoning. We formalize this decision as a small yet expressive *relevance–redundancy optimization* and solve it with a greedy algorithm that enjoys a provable approximation ratio.

Our first step is to choose a subset of sources $\mathcal{A} \subseteq \mathcal{S}$ with the given query $q$. For each source $s \in \mathcal{S}$ we invoke a fixed recall engine[1] and obtain the top-$m$ candidates $\mathcal{C}_s(q) = \{d_{s,1}, \ldots, d_{s,m}\}$. $\mathbf{q} = g_{\text{enc}}(q)$ is the query embedding of $q$. A source representation $\mathbf{s} \in \mathbb{R}^d$ is formed by averaging the frozen document embeddings[2] of its candidates:

$$\mathbf{s} = \frac{1}{m} \sum_{j=1}^{m} \mathbf{d}_{s,j}, \qquad \mathbf{d}_{s,j} = g_{\text{enc}}(d_{s,j}). \tag{1}$$

We define a scoring function $f : 2^{\mathcal{S}} \to \mathbb{R}$ over subsets of sources for a candidate subset $\mathcal{A} \subseteq \mathcal{S}$ as:

$$f(\mathcal{A}; \lambda) = \sum_{s \in \mathcal{A}} \cos(\mathbf{q}, \mathbf{s}) - \lambda * \sum_{s,s' \in \mathcal{A}, s < s'} \cos(\mathbf{s}, \mathbf{s}'), \tag{2}$$

where the first term measures query relevance and the second quantifies inter-source redundancy, and $\lambda \in (0, 1)$ trades off relevance and redundancy. The relevance term favors sources containing content aligned with the query semantics; the redundancy term penalizes sources whose content is highly similar to one another. It can be read as reward the marginally informative, punish the already-covered.

---

[1] Any sparse or dense retriever is admissible; the choice does not affect downstream optimization.
[2] We use sentence-BERT; other encoders yield similar behavior.

We now turn to the sub-modularity of equation equation 2. The first term is modular by construction, as it is a sum of independent contributions. The redundancy term is pairwise and symmetric, and under cosine similarity, it induces a sub-modular structure due to increasing overlap with more sources. Since the sum of a modular and a sub-modular function remains sub-modular, the full objective $f$ is submodular (Carbonell & Goldstein, 1998) when $\lambda$ is small. This structure allows us to employ a greedy algorithm for subset selection with a provable approximation guarantee. We initialize $\mathcal{A}_0 = \varnothing$. For $t = 0, \ldots, |\mathcal{S}| - 1$ we compute the marginal gain $\Delta_t(s) = f(\mathcal{A}_t \cup \{s\}; \lambda) - f(\mathcal{A}_t; \lambda)$ for every $s \in \mathcal{S} \setminus \mathcal{A}_t$ and set $\mathcal{A}_{t+1} = \mathcal{A}_t \cup \{\arg\max_s \Delta_t(s)\}$. The loop stops once $|\mathcal{A}_{t+1}| = |\mathcal{S}|$ or no positive gain exists.

We denote the final set by $\mathcal{A}_{\text{greedy}}$, classical results on submodular (Carbonell & Goldstein, 1998) maximization imply the following guarantee:

$$f(\mathcal{A}_{\text{greedy}}; \lambda) \geq (1 - 1/e) \max_{\mathcal{A} \subseteq \mathcal{S}, |\mathcal{A}| \leq |\mathcal{S}|} f(\mathcal{A}; \lambda). \tag{3}$$

The union of their candidate sets $\mathcal{C}^\star(q) = \bigcup_{s \in \mathcal{A}_{\text{greedy}}} \mathcal{C}_s(q)$ serves as the input to the gradient-aware re-ranking stage. This two-level selection retains complementary evidence while sharply reducing cross-source redundancy.

### 3.3 GRADIENT-AWARE RE-RANKING

Our source selection stage has already removed most irrelevant or duplicated corpora, leaving a mixed candidate pool $\mathcal{C}^\star(q) = \bigcup_{s \in \mathcal{A}^\star} \mathcal{C}_s(q) = \{d_1, ..., d_n\}$. The final bottleneck is the frozen language model's context limit: it can absorb at most $k$ passages. The usual remedy is to train a cross-encoder or to reuse query–document similarity scores, but both approaches ignore how the generator itself reacts to each passage. GRO-RAG therefore lets the LLM "vote": we ask, "If I boost passage $i$, will my loss decrease?"—and we obtain the answer from a single forward–backward pass.

For each candidate document $d_i \in \mathcal{C}^\star(q)$ we already possess the frozen contextual embedding $\mathbf{h}_i$. Selecting exactly $k$ passages is combinatorial. We relax the binary choice (keep / drop) to a soft non-negative weight vector $\pi \in \Delta^n$ ($\pi_i \in [0, 1], \|\pi\|_1 = 1$), we write the mixture representation $\bar{h}(\pi) = \sum_{i=1}^n \pi_i \mathbf{h}_i$. Next, we construct a soft prompt $\langle q, \bar{h}(\boldsymbol{\pi}) \rangle$, and compute the generation loss $\mathcal{L}(\boldsymbol{\pi}) = \mathcal{L}(a^* | q, \bar{h}(\boldsymbol{\pi}))$ with respect to the reference answer $a^*$. Although this relaxation converts the discrete Top-$k$ selection into a continuous problem, the resulting loss function $\mathcal{L}(\cdot)$ remains non-convex and analytically intractable due to the non-linear behavior of the generator. To obtain a tractable approximation, we apply a first-order Taylor expansion around the uniform mixture $\bar{\boldsymbol{\pi}} = (1/n, \ldots, 1/n)$. This yields:

$$\mathcal{L}(\pi) \approx \mathcal{L}(\bar{\pi}) + \langle \nabla_{\bar{h}} \mathcal{L}, \bar{h}(\pi) - \bar{h}(\bar{\pi}) \rangle \tag{4}$$

Considering $\bar{h}(\pi) = \sum_{i=1}^n \pi_i \mathbf{h}_i$ and $\bar{h}(\bar{\pi}) = \frac{1}{n} \sum_{i=1}^n \mathbf{h}_i$, we obtain:

$$\mathcal{L}(\pi) \approx \mathcal{L}(\bar{\pi}) + \sum_{i=1}^n \pi_i \langle \nabla_{\bar{h}} \mathcal{L}, \mathbf{h}_i \rangle \tag{5}$$

Minimizing the loss is therefore approximately equivalent to minimizing a linear weighted sum over document scores. Since $\boldsymbol{\pi}$ is constrained to lie in a $k$-sparse simplex, the approximate optimal solution is obtained by selecting the $k$ documents with the largest negative inner products $\langle \mathbf{h}_i, -\nabla_{\bar{\mathbf{h}}} \mathcal{L} \rangle$. We define the ranking score of document $d_i$ accordingly:

$$\phi_i = \langle \mathbf{h}_i, -\nabla_{\bar{\mathbf{h}}} \mathcal{L} \rangle. \tag{6}$$

This value estimates the sensitivity of the generation loss to the presence of $d_i$ in the prompt, and serves as a posterior-aware signal for document selection. The larger $\phi_i$, the more sharply the loss would drop if document $i$ received additional weight. We keep the $k$ documents with the highest $\phi_i$, set all other weights to zero, and re-normalize. The selected passages $D^\star(q)$ are finally pre-pended to the query. In practice we simply concatenate their raw text, but one could alternatively keep the mixed vector $\bar{h}$ to save context tokens. That is, we only need one forward-backward pass to sort the candidate documents. This strategy requires no additional training, and leverages LLM-internal gradients to estimate document utility with respect to the actual generation objective, not just similarity.

A natural way to measure how much a single document $d_i$ actually helps the generator is to remove it from the context, run the model again, and observe how much the loss increases. We call this

*leave-one-out* (LOO) loss. Formally, starting from the uniform mixture $\bar{\pi} = (\frac{1}{n}, \dots, \frac{1}{n})$ over all $n$ candidates, we define

$$\mathcal{L}_{loo}(d_i) = \underbrace{\mathcal{L}(\bar{\pi})}_{\text{all documents}} - \underbrace{\mathcal{L}\left(\bar{\pi} - \frac{1}{n}\mathbf{e}_i\right)}_{\text{document } i \text{ removed}}, \tag{7}$$

where $\mathbf{e}_i$ is the $i$-th basis vector. Computing this quantity for every passage would require $n+1$ forward passes—prohibitively slow. Below we show that a *single backward pass* yields a gradient score $\phi_i$ that upper-bounds LOO loss, thereby providing a safe ranking surrogate.

**Proposition 3.1 (Gradient inner product upper-bounds leave-one-out)** *Let $g = \nabla_{\bar{h}}\mathcal{L}(\bar{\pi})$ and $\phi_i = \langle \mathbf{h}_i, -g \rangle$. If the one-dimensional function $\ell_i(t) = \mathcal{L}(\bar{\pi} + t\,\mathbf{e}_i)$ is convex on the interval $t \in [-\frac{1}{n}, \varepsilon]$ for some $\varepsilon > 0$, then for every documents $d_i$*

$$\mathcal{L}_{loo}(d_i) \leq -\phi_i.$$

Thus $\phi_i$ upper-bounds the true marginal utility, so ranking by $\phi$ is guaranteed to prioritize passages whose absence would hurt the loss the most.

The gradient-based scoring described above is a single-step forward-backward pass: we linearize the loss landscape around a uniform mixture $\bar{\pi} = (\frac{1}{n}, \dots, \frac{1}{n})$ and select the top-$k$ documents accordingly. Moreover, we can extend this process into a multi-step optimization routine. At each iteration $t$, we maintain a soft weight vector $\pi^t$, use it to form the context $\bar{h}(\pi^t) = \sum_i \pi_i^t \mathbf{h}_i$, and compute the generation loss and its gradient via a forward–backward pass. This yields the descent direction $g^t = \nabla_{\bar{h}}\mathcal{L}(\bar{h}(\pi^t))$, which we use to update $\pi^t$ via gradient descent. The updated weights are then projected back onto the $k$-sparse simplex to maintain feasibility. In effect, this defines an iterative refinement process over document mixtures, allowing the model itself to guide the selection toward increasingly informative subsets. Each iteration re-evaluates the loss under the *current* document mixture and refines the weights accordingly—no new model parameters are introduced, and the only computation is an additional forward-backward pass of the frozen LLM.

**Proposition 3.2 (Linear convergence of the iterative loop)** *Let $\mathcal{L}(\pi) = \mathcal{L}(a^* \mid q, \bar{h}(\pi))$ denote the generation loss evaluated at $\bar{h}(\pi) = \sum_{i=1}^n \pi_i \mathbf{h}_i$. Assume $\mu$-strong convexity and $L$-smoothness of $\mathcal{L}$ in the sub-space $\text{span}\{\mathbf{h}_1, \dots, \mathbf{h}_n\}$. Starting from the uniform vector $\bar{\pi} = (1/n, \dots, 1/n)$, repeat*

$$\tilde{\pi}^{t+1} = \pi^t - \eta\nabla_\pi\mathcal{L}(\pi^t), \qquad \pi^{t+1} = \Pi_{\Delta^{n-1}}(\tilde{\pi}^{t+1}), \qquad 0 < \eta \leq 1/L,$$

*where $\Pi_{\Delta^{n-1}}$ projects onto the probability simplex $\Delta^{n-1} = \{\pi \geq 0, \|\pi\|_1 = 1\}$. Then for all $t \geq 0$:*

$$\mathcal{L}(\pi^{t+1}) - \mathcal{L}^\star \leq (1 - \eta\mu)\left[\mathcal{L}(\pi^t) - \mathcal{L}^\star\right],$$

*with optimal $\mathcal{L}^\star = \min_{\pi \in \Delta^{n-1}} \mathcal{L}(\pi)$. Hence each additional iteration contracts the sub-optimality by the factor $(1 - \eta\mu)$ and therefore never worsens the one-step solution. A proof is provided in Appendix B.*

## 4 EXPERIMENTS

### 4.1 EXPERIMENTAL SETTINGS

**Dataset** Following previous works (Yao et al., 2023; Trivedi et al., 2022a; Zhao et al., 2024b), We evaluate our method on four widely used question answering benchmarks that span both open-domain and multi-hop reasoning settings: **HotpotQA** (Yang et al., 2018), **2WikiMultihopQA** (Ho et al., 2020), and **MuSiQue** (Trivedi et al., 2022b). These datasets each provide a set of ground-truth documents (typically 10-20) for each question as well as ground-truth answers.

**Metrics** We assess model performance from both retrieval and generation perspectives. For retrieval evaluation, we use **nDCG@$k$ (Normalized Discounted Cumulative Gain)** (Burges et al., 2005), which measures the ranking quality of selected documents based on graded relevance and position. A higher nDCG indicates that relevant documents are ranked closer to the top. Ground-truth relevance annotations—where available—are used as supervision. For generation evaluation, we adopt two standard QA metrics (Gao et al., 2023): **Exact Match (EM)**, which measures the proportion of

Table 1: Results (%) of GRO-RAG and baselines on three datasets. "Bold" and "Underlined" denote the highest absolute values and second highest values, respectively.

| LLM | RAG methods | | HotpotQA | | 2WikimQA | | MuSiQue | |
|---|---|---|---|---|---|---|---|---|
| | | | F1 | EM | F1 | EM | F1 | EM |
| Llama3.1-8B | w/o Retrieval | - | 27.8 | 23.1 | 19.7 | 13.9 | 8.4 | 3.5 |
| | Vanilla RAG | Local | 34.2 | 28.2 | 24.1 | 19.4 | 13.1 | 8.4 |
| | | Web | 31.5 | 24.8 | 20.4 | 15.3 | 10.5 | 5.1 |
| | | Both | 36.0 | 29.7 | 27.3 | 21.8 | 15.9 | 9.2 |
| | Single-Source RAG | Self-RAG | 32.3 | 26.4 | 21.1 | 17.4 | 14.8 | 8.9 |
| | | FLARE | 34.5 | 28.6 | 28.5 | 23.0 | 17.3 | 10.7 |
| | | RankRAG | 31.9 | 24.3 | 25.7 | 20.8 | 13.6 | 7.9 |
| | Multi-Source RAG | CRAG | 34.2 | 25.5 | 22.6 | 17.9 | 16.2 | 9.2 |
| | | **GRO-RAG** | 39.1 | 30.9 | 28.9 | 22.8 | 18.6 | 10.3 |
| GLM-4 | w/o Retrieval | - | 29.4 | 23.6 | 18.6 | 13.5 | 10.3 | 4.1 |
| | Vanilla RAG | Local | 36.8 | 29.8 | 25.3 | 20.1 | 13.0 | 8.2 |
| | | Web | 30.4 | 23.9 | 19.5 | 14.8 | 9.4 | 4.3 |
| | | Both | 39.3 | 31.5 | 28.2 | 22.4 | 16.5 | 9.6 |
| | Single-Source RAG | Self-RAG | 34.4 | 28.3 | 22.8 | 18.3 | 17.8 | 10.5 |
| | | FLARE | 38.6 | 30.7 | 29.7 | **23.8** | 20.2 | 11.6 |
| | | RankRAG | 33.2 | 27.3 | 27.4 | 21.6 | 15.8 | 8.8 |
| | Multi-Source RAG | CRAG | 38.1 | 30.3 | 24.8 | 20.4 | 17.4 | 9.6 |
| | | **GRO-RAG** | **42.8** | **33.6** | **30.3** | 23.7 | **21.1** | **12.4** |

generated answers that match ground-truth strings exactly, and **F1 score**, which captures the overlap between predicted and reference answers based on precision and recall. Retrieval metrics are computed over the Top-$k$ ranked documents, while generation metrics are reported over answers produced by the language model conditioned on the selected document set. Here, we set $k = 10$.

**Retrieval settings** To support multi-source retrieval, we leverage both local corpora and web sources. Specifically, for web search, we employ the DuckDuckGo API—a publicly available interface—to access large-scale online information. We compare with three representative retrievers: BM25 (Robertson et al., 2009), a sparse keyword-based method; E5-base (Wang et al., 2022), a dense dual-encoder trained via contrastive learning; and BGE-M3 (Chen et al., 2024), a dense multilingual and multitask-aligned retriever.

**Generation settings** We compare GRO-RAG against four categories of baselines. (1) w/o-Retrieval: LLM directly answers questions without access to any external documents. (2) Vanilla RAG: Standard RAG methods that retrieve documents from either a local corpus, a web corpus, or both, and concatenate them with query as input to the LLM. (3) Single-source RAG: Methods such as Self-RAG (Asai et al., 2023), FLARE (Jiang et al., 2023), and RankRAG (Yu et al., 2024), which rely on a single retrieval source (e.g., local corpus only). (4) Multi-source RAG includes CRAG (Yan et al., 2024) conduct one-time retrieval from the primary source, followed by a one-time supplementary retrieval from a secondary source. For all methods, we conduct experiments based on two built-in LLMs, including Llama3.1-8B (Grattafiori et al., 2024; Meta AI, 2024) and GLM4 (GLM et al., 2024).

## 4.2 MAIN RESULTS

**Performance of Generation** Table 1 presents a comprehensive comparison across three representative QA benchmarks—HotpotQA, 2WikiMQA, and MuSiQue—under both LLaMA3.1-8B and GLM-4 language models. We begin by observing the general trend that retrieval substantially boosts performance over generation-only settings. Across all datasets and models, methods with access to external documents consistently outperform the No-Retrieval baseline, confirming the necessity of knowledge augmentation for multi-hop reasoning tasks. Among vanilla strategies, retrieving from the local corpus tends to be more effective than the web corpus alone, likely due to better domain alignment and lower noise. Concatenating both further improves accuracy, suggesting local and web

Table 2: Comparison of NDCG@10 for different re-ranking methods on three QA datasets using only local corpus. GRO-RAG uses gradient-based re-ranking from frozen LLMs (LLaMA3-8B and GLM-4), while BM25, BGE-M3, and E5-base serve as heuristic or dense retrieval baselines.

| Different rerankers | HotpotQA | 2WikimQA | MuSiQue | Average |
|---|---|---|---|---|
| BM25 | 0.6237 | 0.5760 | 0.3453 | 0.5150 |
| BGE-M3 | 0.6892 | 0.6273 | 0.3922 | 0.5696 |
| E5-base | 0.7013 | 0.6749 | 0.4180 | 0.5981 |
| GRO-RAG with Llama3.1-8B | 0.6442 | 0.6345 | 0.4039 | 0.5609 |
| GRO-RAG with GLM-4 | 0.6538 | 0.6382 | 0.4156 | 0.5692 |

sources offer complementary coverage. However, simply merging top documents from multiple sources is suboptimal. Vanilla RAG (Both) and CRAG, while better than single-source retrieval, rely on static combination rules and lack an understanding of which sources are truly informative. GRO-RAG addresses this through a source combination module that jointly considers query relevance and semantic redundancy, selecting a subset of sources that are diverse and non-overlapping. This leads to improved retrieval precision while avoiding wasted context on near-duplicate content. Compared to adaptive single-source methods such as FLARE and Self-RAG, GRO-RAG consistently achieves stronger performance across all datasets. These methods typically depend on fallback heuristics or self-generated queries, which may help in certain cases but do not explicitly model the final generation objective. In contrast, GRO-RAG performs gradient-based re-ranking directly using the LLM's generation loss, enabling the model to "vote" on which passages are most useful via a single backward pass. This strategy aligns document selection tightly with the downstream objective and proves especially effective under tight context budgets. On more challenging datasets such as MuSiQue, which feature higher document entropy and require more subtle reasoning, the advantage of generation-aware selection becomes even more evident. GRO-RAG not only maintains high accuracy but also demonstrates greater stability across LLMs of varying capacities. For instance, while many baselines suffer performance drops when moving from GLM-4 to the smaller LLaMA3.1-8B, GRO-RAG's relative improvements remain consistent, highlighting its robustness and model-agnostic nature. Lastly, we emphasize that GRO-RAG achieves these gains without any fine-tuning or additional model parameters. All retrieval and scoring steps are conducted with frozen models, making GRO-RAG readily deployable in practical settings.

**Performance of Retrieval**   To ensure fair comparison and isolate the impact of re-ranking, we constrain retrieval to a single local corpus and use identical candidate pools for all methods in a single source setting. The results are shown in Tab 2. We compare GRO-RAG against three common baselines: BM25, a sparse term-matching method; BGE-M3, a modern dense retriever trained with contrastive supervision; and E5-base, a strong general-purpose embedding model. GRO-RAG uses a training-free re-ranking approach that computes gradient-based importance scores from the frozen language model. Despite not using any retrieval supervision or document–query similarity learning, GRO-RAG consistently improves over BM25 and performs on par with or close to strong dense retrievers. In particular, on the most difficult dataset (MuSiQue), GRO-RAG achieves higher nDCG than BGE-M3, suggesting that gradient signals from the generation objective can capture nuanced relevance signals beyond static embeddings. These results demonstrate that even without additional training, GRO-RAG can effectively identify useful documents for the LLM, narrowing the gap to supervised retrievers and providing a principled, efficient alternative for ranking in retrieval-augmented generation. We emphasize that GRO-RAG is not designed to be a standalone retriever; its goal is not to maximize retrieval metrics, but to identify documents most useful for generation. In subsequent experiments, we will show that this strategy leads to consistent improvements in final answer accuracy with minimal computational overhead.

## 4.3 IN-DEPTH ANALYSIS

**Ablation study**   To examine the contribution of each component in GRO-RAG, we conduct an ablation study by removing (1) the Source Combination Selection (SCS) module and (2) the Gradient-aware Re-ranking (GR) module. The results are summarized in Table 3. Removing SCS means retrieving documents independently from all sources without optimizing for relevance–redundancy

Table 3: Ablation study of GRO-RAG. Removing source selection (w/o SCS) or gradient re-ranking (w/o GR) leads to performance degradation across models and datasets.

| LLM | Methods | HotpotQA | | 2WikimQA | | MuSiQue | |
|---|---|---|---|---|---|---|---|
| | | F1 | EM | F1 | EM | F1 | EM |
| Llama3.1-8B | GRO-RAG | **39.1** | **30.9** | **28.9** | **22.8** | **18.6** | **10.3** |
| | w/o SCS | 38.0 | 30.6 | 26.4 | 21.3 | 17.0 | 10.2 |
| | w/o GR | 37.5 | 30.2 | 23.3 | 19.6 | 16.2 | 9.3 |
| GLM-4 | GRO-RAG | **42.8** | **33.6** | **30.3** | **23.7** | **21.1** | **12.4** |
| | w/o SCS | 40.1 | 31.4 | 28.6 | 22.5 | 20.0 | 11.5 |
| | w/o GR | 37.6 | 28.7 | 25.3 | 20.9 | 16.8 | 9.4 |

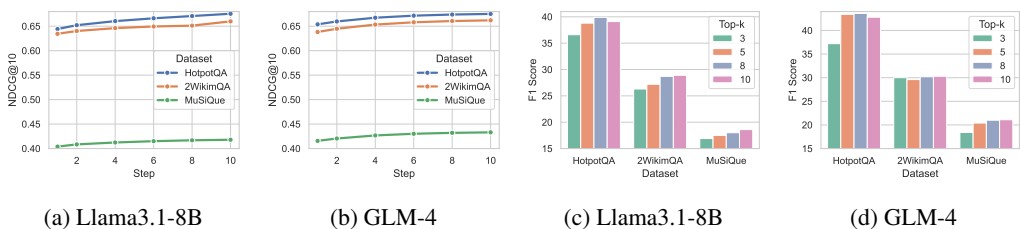

| (a) Llama3.1-8B | (b) GLM-4 | (c) Llama3.1-8B | (d) GLM-4 |

Figure 3: (a)(b) show the effect of optimization steps on retrieval performance (NDCG@10); (c)(d) show the effect of Top-k values on generation performance (F1 score).

tradeoff. While this variant still benefits from having access to multiple corpora, it introduces redundant or semantically overlapping documents into the context, leading to a modest but consistent performance drop across datasets. This suggests that carefully selecting a compact and complementary set of sources helps improve information coverage while minimizing wasteful input. On the other hand, removing GR has a more pronounced effect. This variant replaces gradient-based scoring with a naive per-document loss, ignoring how documents interact when presented jointly. As a result, the model is more likely to prioritize individually informative but contextually redundant documents. Performance declines most noticeably on complex datasets like MuSiQue, indicating that gradient-informed scores better reflect generation utility under constrained context windows.

**Multi-step optimization** Figures 3a and 3b illustrate the NDCG@10 of GRO-RAG under multi-step optimization on three QA datasets using LLaMA3.1 and GLM-4 as backbone LLMs, respectively. In both settings, the curves show consistent and steady improvements with an increasing number of optimization steps. Importantly, the gain in each step diminishes at a near-constant rate, forming a clear linear improvement pattern. This trend empirically supports the theoretical expectation of linear convergence, where the optimization error decreases proportionally across iterations. The convergence behavior is stable across datasets and model scales, with the total gain around (3-5%), validating the efficiency and robustness of GRO-RAG's gradient-based refinement process.

**Impact of different $k$** We investigate how varying the number of retrieved documents ($k$) affects the final answer quality. As shown in Figures 3c and 3d, increasing $k$ does not always lead to better performance. While a larger $k$ provides more information, it may also introduce irrelevant or noisy content, which can interfere with the generation process. We observe that F1 scores generally improve from $k = 3$ to $k = 5$, but further increases sometimes yield diminishing or even negative returns. This highlights importance of carefully selecting $k$ to balance completeness and relevance in RAG.

**Computation times** Computing a single-step gradient using LLaMA3.1-8B takes 882ms on average, while dense retrievers like BGE-M3 complete retrieval in just 39ms. Compared to traditional retrievers, gradient-based retrieval with GRO-RAG introduces significantly higher latency—typically 1 to 2 orders of magnitude slower. This motivates our design choice of using GRO-RAG as a re-ranker, where the initial retrieval already provides a strong prior. In such cases, one gradient step is needed to refine the document selection, making the additional computation cost manageable while still improving answer quality.

## 5 CONCLUSION

In this paper, we introduced GRO-RAG, a training-free and gradient-aware framework for document re-ranking in multi-source Retrieval-Augmented Generation. By directly leveraging generation loss gradients, GRO-RAG identifies Top-$k$ documents that most effectively contribute to the model's output, moving beyond traditional query-based relevance scoring. At the source level, our approach selects complementary sources by jointly modeling query relevance and inter-source redundancy. Theoretically, we show that our re-ranking objective approximates the solution to a loss-minimizing subset selection problem and aligns with minimizing a leave-one-out upper bound.

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

## A    PROOF FOR PROPOSITION 3.1

**Proposition A.1 (Gradient inner product upper-bounds leave-one-out)** *Let* $g = \nabla_{\bar{h}} \mathcal{L}(\bar{\pi})$ *and* $\phi_i = \langle \mathbf{h}_i, -g \rangle$. *If the one-dimensional function* $\ell_i(t) = \mathcal{L}(\bar{\pi} + t\,\mathbf{e}_i)$ *is convex on the interval* $t \in [-\frac{1}{n}, \varepsilon]$ *for some* $\varepsilon > 0$, *then for every documents* $d_i$

$$\mathcal{L}_{loo}(d_i) \leq -\phi_i.$$

Express LOO as a directional difference. From the definition of $\bar{h}(\pi) = \sum_j \pi_j \mathbf{h}_j$:

$$\bar{h}\big(\bar{\pi} - \tfrac{1}{n} e_i\big) = \bar{h}(\bar{\pi}) - \tfrac{1}{n}\mathbf{h}_i.$$

Hence $\mathcal{L}_{\text{loo}}(d_i) = \ell_i(0) - \ell_i(-\frac{1}{n})$.

Apply convexity of $\ell_i$. For a convex function $f$, $f(y) \geq f(x) + f'(x)(y-x)$. Setting $x = 0$, $y = -\frac{1}{n}$ gives

$$\ell_i\big(-\tfrac{1}{n}\big) \geq \ell_i(0) + \ell_i'(0)\big(-\tfrac{1}{n}\big).$$

Rearrange. Subtract the right-hand side from $\ell_i(0)$:

$$\ell_i(0) - \ell_i\big(-\tfrac{1}{n}\big) \leq \tfrac{1}{n}\,\ell_i'(0).$$

Convert the directional derivative. Chain rule yields $\ell_i'(0) = \langle \mathbf{h}_i, \nabla_{\bar{h}} \mathcal{L}(\bar{\pi}) \rangle = \langle \mathbf{h}_i, g \rangle$.

Substitute and flip sign. Therefore

$$\mathcal{L}_{\text{loo}}(d_i) \leq \tfrac{1}{n}\langle \mathbf{h}_i, g \rangle \leq -\phi_i.$$

The factor $\frac{1}{n}$ cancels because it is positive and identical for all passages, leaving the desired inequality.

## B    PROOF FOR PROPOSITION 3.2

**Proposition B.1 (Linear convergence of the iterative loop)** *Let $\mathcal{L}(\pi) = \mathcal{L}\big(a^* \mid q, \bar{h}(\pi)\big)$ denote the generation loss evaluated at $\bar{h}(\pi) = \sum_{i=1}^{n} \pi_i \mathbf{h}_i$. Assume $\mu$-strong convexity and $L$-smoothness of $\mathcal{L}$ in the sub-space $\mathcal{H} = \mathrm{span}\{\mathbf{h}_1, \ldots, \mathbf{h}_n\}$. Starting from the uniform vector $\bar{\pi} = (1/n, \ldots, 1/n)$, repeat*

$$\tilde{\pi}^{t+1} = \pi^t - \eta \nabla_\pi \mathcal{L}(\pi^t), \qquad \pi^{t+1} = \Pi_{\Delta^{n-1}}\big(\tilde{\pi}^{t+1}\big), \qquad 0 < \eta \leq 1/L,$$

*where $\Pi_{\Delta^{n-1}}$ projects onto the probability simplex $\Delta^{n-1} = \{\pi \geq 0, \|\pi\|_1 = 1\}$. Then for all $t \geq 0$:*

$$\mathcal{L}\big(\pi^{t+1}\big) - \mathcal{L}^\star \leq (1 - \eta\mu)\left[\mathcal{L}\big(\pi^t\big) - \mathcal{L}^\star\right],$$

*with optimal $\mathcal{L}^\star = \min_{\pi \in \Delta^{n-1}} \mathcal{L}(\pi)$. Hence each additional iteration contracts the sub-optimality by the factor $(1 - \eta\mu)$ and therefore never worsens the one-step solution.*

**Proof**:

We have:

$$\Delta^{n-1} = \{\pi \in \mathbb{R}^n \mid \pi_i \geq 0, \|\pi\|_1 = 1\}, \qquad \Pi_{\Delta^{n-1}}(v) = \arg\min_{\pi \in \Delta^{n-1}} \|\pi - v\|_2. \tag{8}$$

For any iterate $\pi^t$ define the mixed-context hidden vector $\bar{h}(\pi^t) = \sum_i \pi_i^t \mathbf{h}_i$ and let $G^t = \nabla_\pi \mathcal{L}(\pi^t)$. Because $\bar{h}(\pi) = M\pi$ with $M = [\mathbf{h}_1, \ldots, \mathbf{h}_n]$, the chain rule gives

$$G^t = M^\top \nabla_{\bar{h}} \mathcal{L}\big(\pi^t\big). \tag{9}$$

$\mathcal{L}$ is $L$-smooth on $\mathcal{H}$, hence for any $\pi, v \in \mathbb{R}^n$:

$$\mathcal{L}(v) \leq \mathcal{L}(\pi) + \langle \nabla_\pi \mathcal{L}(\pi), v - \pi \rangle + \frac{L}{2} \|v - \pi\|_2^2. \tag{10}$$

Apply equation 10 with $\pi = \pi^t$ and $v = \tilde{\pi}^{t+1} = \pi^t - \eta G^t$:

$$\mathcal{L}(\tilde{\pi}^{t+1}) \leq \mathcal{L}(\pi^t) - \eta \langle G^t, G^t \rangle + \frac{L\eta^2}{2} \|G^t\|_2^2 = \mathcal{L}(\pi^t) - \eta\Big(1 - \tfrac{L\eta}{2}\Big)\|G^t\|_2^2. \tag{11}$$

Let $\pi^\star$ be a minimizer of $\mathcal{L}$ on $\Delta^{n-1}$. Using the Pythagorean property of Euclidean projection,

$$\|\pi^{t+1} - \pi^\star\|_2^2 \leq \|\tilde{\pi}^{t+1} - \pi^\star\|_2^2. \tag{12}$$

Because $\mathcal{L}$ is $\mu$-strongly convex,

$$\mathcal{L}(\pi) - \mathcal{L}(\pi^\star) \geq \frac{\mu}{2} \|\pi - \pi^\star\|_2^2, \qquad \forall \pi \in \Delta^{n-1}. \tag{13}$$

Combine equation 12 and equation 13 to get

$$\mathcal{L}(\pi^{t+1}) - \mathcal{L}(\pi^\star) \leq \mathcal{L}(\tilde{\pi}^{t+1}) - \mathcal{L}(\pi^\star). \tag{14}$$

Another consequence of strong convexity is

$$\|G^t\|_2^2 \geq 2\mu \left[\mathcal{L}(\pi^t) - \mathcal{L}(\pi^\star)\right]. \tag{15}$$

Insert equation 15 into the descent lemma equation 11 and use $0 < \eta \leq 1/L \Rightarrow 1 - \frac{L\eta}{2} \geq \frac{1}{2}$:

$$\mathcal{L}(\tilde{\pi}^{t+1}) \leq \mathcal{L}(\pi^t) - \eta\mu\left[\mathcal{L}(\pi^t) - \mathcal{L}(\pi^\star)\right]. \tag{16}$$

Subtract $\mathcal{L}(\pi^\star)$ from both sides and combine with equation 14:

$$\mathcal{L}(\pi^{t+1}) - \mathcal{L}(\pi^\star) \leq (1 - \eta\mu)\left[\mathcal{L}(\pi^t) - \mathcal{L}(\pi^\star)\right]. \tag{17}$$

This proves the claimed inequality with $\mathcal{L}^\star = \mathcal{L}(\pi^\star)$. Iterating the contraction yields

$$\mathcal{L}(\pi^t) - \mathcal{L}^\star \leq (1 - \eta\mu)^t \left[\mathcal{L}(\pi^t) - \mathcal{L}^\star\right], \tag{18}$$

which is geometric linear convergence with rate $1 - \eta\mu$ $(< 1)$.

## C  LIMITATIONS

While GRO-RAG offers a lightweight and training-free alternative for improving document selection in multi-source RAG, it also presents several limitations. First, the gradient-based re-ranking procedure requires a backward pass through the frozen LLM, which, although computationally modest compared to fine-tuning, still introduces latency compared to purely retrieval-side methods. Second, GRO-RAG's ranking relies on local linear approximations of the generation loss, which may become inaccurate when the true loss landscape is highly non-linear or when documents exhibit strong interaction effects. Finally, our source selection module is limited to fixed document embeddings; future extensions could explore query-aware or adaptive source encoders to further enhance selection.

## D  BROADER IMPACTS

GRO-RAG contributes to the growing field of retrieval-augmented generation by proposing a more interpretable, modular, and training-free alternative to dense scoring models. Its emphasis on gradient-based document utility aligns document selection directly with the language model's generation behavior, offering greater transparency and controllability in the retrieval pipeline. By avoiding task-specific fine-tuning, GRO-RAG lowers the resource barriers to deploying RAG systems in practical applications, particularly in low-resource or privacy-sensitive settings.

