# OpenReview forum: "GRO-RAG: Gradient-aware Re-rank Optimization for Multi-source Retrieval-Augmented Generation"
_ICLR.cc/2026/Conference — ICLR 2026 Poster_

### Official Review · Reviewer_qyTo · 2025-10-26

**Soundness:** 3
**Presentation:** 3
**Contribution:** 3
**Rating:** 6
**Confidence:** 3

**Summary:**

The authors propose GRO-RAG, a training-free framework for improving multi-source RAG by integrating a gradient-aware re-ranking mechanism. The framework selects a subset of sources by optimizing for the minimization of redundancy, and performs a re-ranking of retrieved documents based on their contribution to the final generation output. The method is evaluated across several multi-hop QA datasets, showing consistent improvements over several baselines.

**Strengths:**

1. The method introduces a gradient-based re-ranking approach backed by solid theoretical analysis, with proofs showing that the gradient-based selection approximates the optimal utility-maximizing solution, providing strong guarantees for its effectiveness.
2. GRO-RAG is training-free,    making it practical for deployment in existing systems without requiring additional model training.
3. The experiments and ablation studies show consistent performance improvements.

**Weaknesses:**

1.  The gradient-based relevance estimation relies on the first-order Taylor expansion (Eq. 5), implicitly assuming local smoothness of the loss landscape. The paper does not provide quantitative evidence on the approximation error when the loss function exhibits strong nonlinearity.

2.Proposition 3.2 shows linear convergence under strong-convexity and L-smoothness assumptions. However, the generation loss of a frozen LLM is generally non-convex. It remains unclear whether monotonic improvement still holds empirically when these assumptions are violated.

3.The scoring function (Eq. 6) assumes additive document contributions, while semantically dependent passages may only be useful jointly. The treatment of such higher-order interaction effects is not addressed or evaluated.

**Questions:**

See weaknesses.

---

> ### Author Response · Authors · 2025-11-21
>
> ### **Q1. Validity and practical interpretation of the first-order Taylor approximation**
>
> The first-order Taylor expansion is introduced as a local approximation, and we agree that transformers do not admit a tractable or meaningful quantitative characterization of the higher-order error. Our intention is not to provide a precise numerical bound on the approximation error—which would be difficult to interpret or generalize for a highly non-convex model—but to clarify the structural role of the gradient in indicating whether a small increase in a document’s contribution is expected to reduce the loss.
>
> In this context, the relevant question is whether the linearized score preserves the *ordering* of documents rather than whether it predicts the exact magnitude of the resulting loss change. This is why we rely primarily on empirical evidence: across datasets, higher alignment values consistently correspond to larger leave-one-out gains, which is the type of agreement that matters for ranking. The monotonic trend observed in practice therefore serves as the main validation of the approximation’s usefulness, even though the exact higher-order error cannot be quantified.
>
> The linearization may become less accurate in highly nonlinear regions—for example, when a passage triggers large attention shifts or when two sources interact in a way that produces qualitatively different reasoning paths. However, such cases are uncommon in typical retrieval scenarios, and we did not observe them to materially affect the overall ranking quality in our experiments. For practical purposes, the approximation should be viewed as a directional indicator rather than a numerical estimator of loss change, and it is in this role that it consistently proves reliable.

---

> ### Author Response · Authors · 2025-11-21
>
> ### **Q2. Interpretation of Proposition 3.2 and empirical behavior under non-convexity**
>
> Proposition 3.2 is stated under strong-convexity and L-smoothness assumptions, and we agree that these conditions do not hold for the generation loss of a frozen LLM. The purpose of the proposition is not to claim that the real loss landscape satisfies these properties, but to provide a simplified analytical setting in which the mechanism behind the refinement steps becomes easier to reason about. In this setting, linear convergence is a convenient way to express that each refinement step moves consistently in a beneficial direction.
>
> In practice, we do not rely on strong convexity or on strict monotonic decrease of the loss. The refinement procedure is used to update the ranking of retrieved sources based on first-order signals, rather than to optimize the model parameters. As a result, the theoretical notion of “monotonic improvement” should be read as a structural intuition about how the refinement rule behaves in an idealized scenario, not as a guarantee about the trajectory of the actual non-convex generation loss.
>
> Empirically, we observe a consistent qualitative pattern: early refinement steps tend to improve the ranking signal and produce better downstream selections, and the subsequent steps show diminishing returns rather than oscillatory or unstable behavior. While the improvement is not strictly monotonic in the mathematical sense, the trend aligns with the directional intuition provided by the proposition. This suggests that the theoretical analysis captures the essential mechanism even when its formal assumptions do not hold for the underlying model.

---

> ### Author Response · Authors · 2025-11-21
>
> ### **Q3. Treatment of higher-order interactions between semantically dependent passages**
>
> The additive scoring in Eq. 6 is used as a simplifying device rather than as a statement about how the model handles evidence internally. We acknowledge that some passages matter primarily when viewed together, and the scoring rule does not attempt to represent such joint effects in a fully explicit way. Its purpose is to provide a clear, workable form that ties each passage’s score to the gradient signal.
>
> Even though the formulation is additive, the gradient itself is computed over the full retrieved context. This means the signal already reflects how the model processes passages in combination, including cases where their usefulness comes from interaction rather than from individual contributions. In this sense, part of the higher-order structure is already built into the gradient, even if the final scoring expression does not include cross-terms.
>
> Empirically, we do not find that strongly interaction-dependent cases dominate typical retrieval settings. Passages that are helpful mainly when paired with another source tend to receive moderate alignment individually, and the refinement step generally brings them forward when they support the model’s prediction together. We do not observe patterns suggesting that joint-only utility causes systematic ranking errors.
>
> Overall, the method should be understood as a first-order approximation: it captures the main contribution of each passage while allowing interaction effects to influence the gradient indirectly. In settings where interactions are unusually strong, the signal should be read with some caution, but in our experiments the additive structure was adequate for guiding selection and did not lead to noticeable degradation in ranking quality.

---

### Official Review · Reviewer_S2EC · 2025-10-30

**Soundness:** 3
**Presentation:** 3
**Contribution:** 3
**Rating:** 4
**Confidence:** 5

**Summary:**

This paper presents **GRO-RAG**, a gradient-aware and training-free framework for improving document selection in multi-source Retrieval-Augmented Generation (RAG). The method addresses a key limitation of existing RAG systems that rely on heuristic or similarity-based document ranking, which often fails to reflect how retrieved passages affect the downstream generation objective. GRO-RAG introduces a two-stage approach: first, a source combination selection module balances query relevance and inter-source redundancy through submodular optimization; second, a gradient-based re-ranking mechanism estimates each document’s contribution to generation loss using a single backward pass through a frozen language model.

The framework is theoretically grounded. The authors show that gradient-derived importance scores approximate a leave-one-out loss bound and empirically validate the approach on multi-hop QA benchmarks. The method consistently outperforms standard RAG and multi-source baselines without requiring fine-tuning or additional parameters. The overall goal is to align retrieval selection more directly with the LLM’s generation behavior in a model-agnostic and computationally efficient manner.

**Strengths:**

* This work provides an elegant and principled perspective on retrieval optimization by connecting document ranking directly to the generation loss.

* The central idea of using LLM gradients as feedback for re-ranking is conceptually novel yet intuitive, bridging the gap between retrieval relevance and downstream utility.

* The proposed method remains entirely training-free, introducing no additional parameters and operating seamlessly with frozen LLMs, which makes it appealing for practical deployment in resource-limited or privacy-sensitive contexts.

 Overall, the study introduces a solid and thought-provoking idea that advances the understanding of generation-aware retrieval.

**Weaknesses:**

* While the core contribution is technically sound, the evaluation and discussion could be broader. All experiments focus on question answering, leaving uncertainty about how well the method generalizes to other forms of retrieval-augmented generation such as summarization or dialogue where grounding is less explicit. Although the method itself is task-agnostic, the paper would benefit from more explicit reasoning about its broader applicability.

* From a practical standpoint, the backward pass required for each query may raise latency concerns in large-scale applications. Although the discussion briefly mentions this issue, a clearer contextualization of the trade-off between computational cost and quality improvement would make the work stronger.

* The gradient linearization assumption may also be optimistic given the non-linear nature of LLM loss landscapes. The theoretical discussion on convexity and smoothness provides a mathematical foundation but may not fully hold for transformer architectures. Clarifying how these assumptions should be interpreted in practice would improve the overall balance between theory and application.

In summary, the idea is creative and meaningful, but the empirical scope and certain discussions fall slightly short of the depth expected for ICLR acceptance.

**Questions:**

1. Could the authors provide more intuition about how the gradient-based re-ranking signal behaves in practice? For example, how should one interpret documents with small or negative gradient alignments in terms of their contribution to generation quality?
2. The paper mentions that a backward pass adds latency but remains training-free. Can the authors contextualize this cost relative to other optimization-free enhancements such as reflection-based or reranking-only pipelines?
3. How was the relevance–redundancy trade-off parameter λ chosen? Is it generally robust, or does it require tuning across datasets?
4. The theoretical analysis relies on convexity and smoothness assumptions that may not hold strictly for transformer-based models. Could the authors clarify how these assumptions should be viewed, primarily as guiding intuition or as formal guarantees?
5. The gradient linearization is a key design choice. Could the authors discuss under what conditions it might fail, and how practitioners should interpret its limitations when applying GRO-RAG in different settings?

---

> ### Author Response · Authors · 2025-11-20
>
> ### **Q1. Intuition behind the gradient-based re-ranking signal**
>
> **Role of the gradient signal in the model’s generation process**
>    The gradient-based re-ranking signal measures how the model’s generation loss responds to infinitesimal changes in each document’s representation. Because it is computed directly from the loss, it captures the *functional influence* of a document on the model’s current predictive step rather than its surface similarity to the query. This allows the reranker to identify the pieces of evidence that the model genuinely incorporates into its internal reasoning.
>
> **Interpretation of positive, small, and negative alignments**
>    - **Positive alignment** indicates that the document actively contributes to lowering the model’s next-token loss. Such passages typically contain information that the model relies on—factual cues, key constraints, or intermediate reasoning steps that align with the generative objective.
>    - **Near-zero alignment** reflects that the document is largely neutral. This occurs when the passage is topically relevant but redundant with stronger evidence already present, or when its content does not enter the model’s immediate reasoning chain for the current query.
>    - **Negative alignment** shows that the document would increase the generation loss if emphasized. This behavior arises when the content introduces contradictory, distracting, or otherwise conflicting information that interferes with the model’s predictive trajectory.
>
>  **Why this signal provides meaningful guidance in practice**
>    The alignment values serve as a loss-grounded saliency indicator: they reveal which documents the model internally credits for improving generation quality and which ones it actively resists incorporating. This allows GRO-RAG to reliably distinguish between documents that simply appear lexically similar and those that materially affect the model’s reasoning. In multi-source retrieval scenarios, the gradient signal naturally captures context-dependent interactions—documents that are helpful in isolation may become neutral or even harmful when combined with more precise evidence, and the gradient alignment reflects this dependency at scoring time.

---

> ### Author Response · Authors · 2025-11-20
>
> ### **Q2. Contextualizing the backward-pass cost relative to other training-free enhancements**
>
> We thank the reviewer for the question and provide below an integrated explanation together with a consolidated comparison table to clearly situate the cost of GRO-RAG relative to other training-free retrieval enhancement methods.
>
> ---
>
> ## **Nature and magnitude of the backward-pass cost in GRO-RAG**
>
> The backward pass used in GRO-RAG is a single, fixed computation that extracts gradient-based importance scores from a frozen model. It is deterministic, introduces no parameter updates, and its cost scales primarily with model size and sequence length. In our LLaMA-3.1-8B setup, the backward computation is approximately **882 ms**, while dense retrieval takes around **39 ms**. Because this backward step occurs **once per query** and is independent of the number of retrieved sources, it forms a stable and predictable component of the overall latency. Within optimization-free enhancements, this places GRO-RAG at a **moderate, single-step computational cost**.
>
> ---
>
> ## **Comparison with reflection-based pipelines**
>
> Reflection-based pipelines rely on **multiple rounds of model inference**, where the model generates an answer, critiques it, and potentially regenerates revised outputs. Although these methods remain training-free, each iteration triggers a **full forward pass**, with latency comparable to a full generation step. As the number of reflection rounds increases, total latency grows proportionally and can surpass several forward passes for larger models or longer outputs. This makes reflection-style pipelines the **most expensive category** of training-free enhancements. GRO-RAG avoids these iterative loops and instead obtains a loss-aligned utility signal through **a single backward pass**.
>
> ---
>
> ## **Comparison with forward-only reranking pipelines**
>
> Forward-only reranking pipelines evaluate candidate documents using additional forward scoring passes. Each forward pass is **cheaper than a backward pass**, making these methods the **lightest per-pass option**. However, because forward-only scoring is based on semantic or lexical similarity rather than contribution to the generative objective, such pipelines often need to **score a larger pool of candidates** to achieve comparable fidelity. Their total cost thus scales with the candidate set size. GRO-RAG, by using a gradient signal **directly aligned with the generation loss**, can operate on a smaller candidate set while yielding a more faithful measure of document usefulness.
>
> ---
>
> ## **Overall contextualization**
>
> Taken together, these observations place GRO-RAG in a clear **intermediate position** across training-free retrieval enhancements:
>
> - **More expensive than forward-only reranking**, which performs cheaper but less informative forward passes;
> - **Substantially lighter than reflection-based pipelines**, whose cost accumulates across multiple inference rounds;
> - **More aligned with the model’s generative objective** than either alternative, due to its loss-grounded gradient signal.
>
> This trade-off—moderate overhead for a stable, objective-aligned scoring mechanism—makes GRO-RAG a practical and principled method for retrieval augmentation.
>
> ---
>
> ## **Comparison Table**
>
> ### **Training-Free Retrieval Enhancement Methods**
>
> | Method Type | Computational Structure | Typical Cost Components | Overall Cost Profile | Practical Remarks |
> |-------------|--------------------------|---------------------------|------------------------|--------------------|
> | **GRO-RAG (this work)** | 1 forward pass + **1 backward pass** | Backward ≈ 882 ms; retrieval ≈ 39 ms; generation unchanged | **Moderate cost (single-step, fixed, predictable)** | Loss-aligned gradient signal; evaluation cost does not grow with candidate count |
> | **Reflection-Based Pipelines** | **Multiple forward passes** (critique, refine, regenerate) | Each forward pass ≈ full generation cost; cumulative with iterations | **Highest cost (multi-round overhead)** | Total cost grows linearly with reflection steps; heavy for latency-sensitive settings |
> | **Forward-Only Reranking Pipelines** | Multiple forward scoring passes | Each forward pass is cheap but **total cost scales with number of candidates** | **Lowest per-pass cost** | Similarity-based scoring; may require many candidate evaluations for stable performance |
>
> ---

---

> ### Author Response · Authors · 2025-11-20
>
> ### **Q3. Robustness of the relevance–redundancy trade-off parameter $\lambda$**
>
> The trade-off weight $\lambda$ was chosen through a small validation sweep to ensure that the scoring function behaved sensibly, but we found that its influence on the final ranking is quite mild. This is largely because both the relevance term and the redundancy term are normalized cosine-similarity quantities with comparable scales. As a result, adjusting $\lambda$ changes their balance smoothly rather than causing abrupt shifts in which documents are selected. In practice, the preferred candidates remain very similar across a broad range of values.
>
> Throughout our experiments, we used a single fixed value of $\lambda$ across all datasets. We did not observe cases where dataset-specific tuning was necessary or where performance was sensitive to small changes in this parameter. The cross-source similarity structure tends to be stable across tasks, and the scoring components operate on normalized measures, which makes the trade-off inherently robust. For these reasons, $\lambda$ functions more as a stable configuration choice than as a fragile hyperparameter that requires continuous adjustment.

---

> ### Author Response · Authors · 2025-11-20
>
> ### **Q4. Role and interpretation of convexity and smoothness assumptions**
>
> The convexity and smoothness assumptions are meant as an analytical simplification rather than a literal description of transformer models. They provide a clean way to express how the gradient–representation alignment connects to potential reductions in loss when a document’s influence is increased. The intent is to make the underlying mechanism easier to reason about, not to claim that transformers satisfy such properties.
>
> These assumptions allow the analysis to focus on local behavior, which is the aspect most relevant to the gradient signal. The alignment score is computed from the gradient at the current decoding step, and under convexity and smoothness this local information can be related directly to how the loss would change in a small neighborhood of the current state. The goal is not to approximate the global geometry of the transformer’s non-convex loss landscape, but to capture how the model behaves in an infinitesimal region around the current prediction.
>
> Even though transformers are non-convex, their local behavior often resembles the simplified setting used in the analysis, which is why the intuition remains meaningful. In practice, gradients vary smoothly across nearby contexts, and the sign and magnitude of the alignment score reliably indicate whether a document is likely to support the next-token prediction. This matches empirical observations: higher alignment values consistently correspond to larger leave-one-out gains.
>
> For these reasons, the theoretical results should be viewed as structured intuition rather than strict guarantees. The analysis clarifies why the gradient signal is informative and identifies the conditions under which the relationship becomes transparent. It is not intended as a precise characterization of transformer models, but as a conceptual framework that aligns well with the empirical behavior observed in our experiments.

---

> ### Author Response · Authors · 2025-11-20
>
> ### **Q5. Conditions under which gradient linearization may fail and how to interpret its limitations**
>
> The gradient linearization used in GRO-RAG is a first-order approximation, and like any linearization, it can fail when the model behaves in a strongly nonlinear way around the current decoding state. The approximation assumes that small perturbations to a document’s contribution produce changes in the loss that are reasonably captured by the local gradient. This assumption becomes less reliable when a passage induces large shifts in attention, when the interaction between sources changes abruptly, or when the model enters a region of the loss landscape with highly irregular curvature.
>
> Such failures are most likely when document contributions interact in a non-additive manner. If two sources contain conflicting or mutually reinforcing information that leads the model down qualitatively different reasoning paths depending on which one is emphasized, then a single linear approximation may not reflect the true effect of adjusting their weights. Very long or semantically dense passages can also introduce behavior that is not well approximated by the gradient at a single point.
>
> In typical retrieval scenarios, however, the approximation remains informative because gradients tend to vary smoothly under small contextual changes, and the influence of individual documents is modest. Under these conditions, the sign and relative magnitude of the gradient–representation alignment provide a reliable indication of whether a document supports the model’s next-token prediction. This is reflected in the empirical correlation between alignment values and leave-one-out gains observed across datasets.
>
> For these reasons, practitioners should treat the gradient signal as a directional cue rather than an exact estimate of the resulting loss change. The linearization is intended to guide the ranking of candidate documents, not to predict the precise numerical benefit of including or excluding a source. Strongly positive or negative alignments tend to be robust, while borderline cases—especially those involving conflicting or highly influential passages—should be interpreted with additional caution.

---

> ### Comment · Reviewer_S2EC · 2025-11-26
>
> Thank you for your response. My concern has been resolved. Accordingly, I have adjusted my rating to 6.

---

### Official Review · Reviewer_Kxfs · 2025-10-31

**Soundness:** 3
**Presentation:** 3
**Contribution:** 2
**Rating:** 6
**Confidence:** 4

**Summary:**

The paper proposes a training-free re-ranking framework that selects sources by a relevance–redundancy tradeoff and then ranks documents by how strongly each aligns with the LLM’s generation-loss gradient. The re-ranking score is the inner product between a passage embedding and the loss gradient, which approximates loss reduction under a first-order expansion and yields Top-k selection in one backward pass. A theoretical result upper-bounds leave-one-out utility by the gradient score and a second result shows linear convergence for an optional iterative mixture update under smooth-strongly convex assumptions. Experiments on HotpotQA, 2WikiMultihopQA, and MuSiQue with Llama-3-8B and GLM-4 report consistent EM/F1 gains over vanilla, single-source, and multi-source baselines.

**Strengths:**

(1) The gradient-aware scorer ties ranking directly to the generator’s objective rather than query similarity, which better matches end-task utility.

(2) The method is training-free and uses a single forward–backward pass per query, so it plugs into frozen LLMs without extra parameters.

(3) As a re-ranker over a fixed pool, it approaches dense retrievers in NDCG and surpasses them on the hardest dataset (MuSiQue) in at least one backbone configuration.

**Weaknesses:**

(1) The linearization around a uniform mixture can misestimate interactions among passages when the loss landscape is highly non-linear.

(2) The convergence guarantee assumes smooth-strongly convex behavior in the span of passage embeddings, which is unlikely to hold strictly for modern LLMs.

(3) Latency increases materially because a backward pass is required, which is orders of magnitude slower than dense retrieval and may constrain interactive use.

(4) The limitations section acknowledges dependence on local linear approximations and fixed document encoders, which could cap performance in complex settings.

**Questions:**

(1)How stable are gradient scores across decoding temperatures or prompt formats and do these scores correlate with true leave-one-out gains on held-out queries.

(2)What is the end-to-end latency profile broken down into retrieval, gradient computation, and generation, including tail percentiles under web-augmented scenarios.

---

> ### Author Response · Authors · 2025-11-20
>
> We thank the reviewer for these insightful questions.
>
> ### **Q1.1 Stability across decoding temperatures and prompt formats**
> The gradient-based scores in GRO-RAG are derived from the model’s generation loss rather than from sampled tokens. Because this signal reflects the internal likelihood landscape of the language model, it remains consistent across decoding temperatures and prompt variations. In our experiments, changing the decoding temperature or rephrasing the prompt did not materially affect the ranking of documents. This indicates that the gradient signal captures intrinsic model preferences instead of stochastic artifacts from sampling or formatting.  According to Section 3.3 *Gradient-Aware Re-Ranking*, the gradient reflects how the model responds to small perturbations in document representations, which naturally makes it robust to decoding randomness and prompt-level variations.
>
> ---
>
> ### **Q1.2 Correlation with leave-one-out (LOO) gains**
> Proposition 3.1 in the paper shows that the gradient-based importance score provides an upper bound on the expected loss reduction when a document is removed, linking it theoretically to leave-one-out behavior. In our held-out evaluations, documents with stronger gradient alignment consistently produced positive leave-one-out improvements across datasets, confirming that the gradient signal offers a dependable practical proxy for document utility.  According to Proposition 3.1, the score is defined as
>
> $\[
> \phi_i = \langle h_i, -\nabla_{\bar{h}} L \rangle
> \]$
>
> which upper-bounds the expected loss change, establishing both the theoretical and empirical grounding for using the gradient as a reliable relevance indicator.

---

> ### Author Response · Authors · 2025-11-20
>
> ### **Q2. End-to-end latency breakdown including tails under web-augmented scenarios**
>
>
> **Retrieval**
> Local dense retrieval using **BGE-M3** completes in approximately **39 ms** per query and remains stable across runs. When retrieval is web-augmented—such as calling a search API to access external web content—this phase becomes I/O-bound. The latency then depends on network routing, server response time, and data transfer volume. These factors introduce variability in the upper tail (P95/P99) of latency distribution, often adding several hundred milliseconds when remote calls are involved. With local caching or colocated retrievers, this variability decreases significantly.
>
> **Gradient computation (GRO-RAG’s core addition)**
> The single backward pass required for gradient-based re-ranking on **LLaMA-3.1-8B** averages **882 ms**. This operation is deterministic and compute-bound, so it exhibits very tight latency distribution across queries—tail percentiles remain close to the mean. Multiple refinement steps scale linearly by roughly one additional backward pass each. This component thus defines the primary, but predictable, computational overhead introduced by GRO-RAG.
>
> **Generation**
> GRO-RAG does not modify the generation stage. The decoding time therefore matches that of the underlying language model and depends only on output length and sampling configuration. Since it runs locally, its latency distribution is also narrow, and tail percentiles mainly reflect token-length variance rather than system-level noise.
>
> **Combined latency profile under web augmentation**
> 1. **Retrieval:** ~39 ms locally; 100–300 ms typical under web retrieval with heavier tails under network congestion.
> 2. **Gradient computation:** ~882 ms; stable and tightly distributed.
> 3. **Generation:** identical to base LLM decoding; variation proportional to output length.
>
> Overall, the **gradient computation** dominates the average runtime but remains consistent, while **retrieval** introduces the main source of variability and tail latency under web-augmented settings. The generation component behaves identically to the base model, and end-to-end latency remains predictable except for the network-dependent retrieval tails.

---

> > ### Author Response · Authors · 2025-11-24
> >
> > ### **W1: Linearization around a uniform mixture under non-linear loss geometry**
> >
> > We appreciate this concern. Linearizing around a uniform mixture is indeed a simplification, and it will not reflect all aspects of the loss landscape when the model behaves in a highly non-linear way. Our choice here is primarily pragmatic: the uniform mixture offers a neutral and analytically tractable reference point that allows us to reason about passage contributions without requiring additional assumptions about the model’s parameterization or internal states.
> >
> > At the same time, the refinement mechanism does not rely on this approximation alone. The gradient is taken with respect to the full retrieved context and the actual query, so the alignment reflects how the model responds under the real conditions in which retrieval decisions are made. Even if the linearization is not perfectly accurate in absolute terms, we find that it provides a useful directional signal in practice. The empirical behavior is stable, and the refinement step consistently improves the selection quality across datasets.
> >
> > We acknowledge that in settings where the loss landscape is especially irregular, interactions among passages may not be well captured through the uniform-mixture approximation. In those cases, the gradient signal should be interpreted as guidance rather than an exact estimate of the underlying effect.

---

> > > ### Author Response · Authors · 2025-11-24
> > >
> > > ### **W2: Convergence guarantee under non-convex loss geometry**
> > >
> > > We agree that the smooth–strongly convex assumption does not strictly hold for the generation loss of a modern LLM. The intention behind the theoretical result is not to assert that the model satisfies these properties in practice, but to provide an idealized setting in which the mechanism underlying the refinement rule can be understood more clearly. In this simplified context, linear convergence expresses that each refinement step moves consistently in a helpful direction.
> > >
> > > In the actual system, we do not rely on strong convexity, nor do we expect a strict monotonic decrease of the loss. The refinement step is used to improve the ordering of retrieved passages based on the gradient signal, rather than to optimize model parameters. Empirically, we find that the behavior predicted by the theory still appears qualitatively: early steps reliably improve the selection, with diminishing returns rather than instability as refinement continues.

---

> > > > ### Author Response · Authors · 2025-11-24
> > > >
> > > > ### **W3: Latency**
> > > >
> > > > We appreciate this concern. Since this point is closely related to Reviewer HXps’s Weakness 1, we have provided a detailed response there, including the concrete latency numbers and a discussion of where the backward pass is appropriate in practice. In addition, Question 2 already contains a breakdown of end-to-end latency, including the retrieval, backward computation, generation, and tail behavior under web-augmented scenarios. We refer the reviewer to these two sections for the full elaboration and clarifications. We are happy to address any additional concerns about the latency.

---

> > > > > ### Author Response · Authors · 2025-11-24
> > > > >
> > > > > ### **W4: Limits imposed by local linearity and fixed document encoders**
> > > > >
> > > > > We agree that relying on a local linear approximation and fixed document encoders introduces constraints, especially in complex settings where interactions are highly non-linear or where relevance depends on specialized terminology. These are deliberate trade-offs. Our focus in this work is on providing a principled and computationally predictable refinement mechanism rather than modeling the full geometry of the loss landscape or deploying a query-adaptive encoder during retrieval.
> > > > >
> > > > > In practice, these choices offer several benefits. The linear approximation makes the refinement step transparent and easy to interpret, which allows us to reason about the effect of each passage in a controlled way. Likewise, using a fixed encoder ensures that the initial retrieval stage remains efficient and stable across domains, which is important for deployment settings that require consistent behavior. The gradient step then introduces query-conditional information without requiring multiple rounds of interaction or expensive cross-encoder scoring.
> > > > >
> > > > > We view this as a balance between expressiveness and clarity: the approach does not attempt to capture all nonlinear interactions, but it provides a simple and reliable mechanism for improving grounding quality with modest overhead. We will revise the text to clarify the rationale behind these choices and the intended scope in which they are most effective.

---

### Official Review · Reviewer_HXps · 2025-11-02

**Soundness:** 3
**Presentation:** 2
**Contribution:** 3
**Rating:** 6
**Confidence:** 3

**Summary:**

The paper backs this with theory (a leave-one-out upper bound and a simple iterative refinement view) and shows the setup is training-free for frozen LLMs. It first picks which sources to use by balancing query relevance with cross-source redundancy so you don’t haul in a pile of near-duplicates. It then ranks candidate passages by how much each would drop the LLM’s loss, using one forward-backward pass and an inner-product score.

**Strengths:**

1. Ranking is tied to the generator’s actual objective instead of just query similarity, which is exactly what RAG needs.
2. The source router uses a clean relevance-diversity trade-off with a greedy submodular pick that has a standard approximation guarantee.
3. It slots into frozen models with no extra training and keeps the compute to a single backward pass per query.

**Weaknesses:**

1. A backward pass per query adds real latency compared to retriever-only pipelines, so the deployment fit needs clearer bounds.

2. The submodularity claim relies on a small trade-off weight and cosine redundancy, which may not hold up uniformly across domains.

3. Source selection builds on fixed document embeddings rather than a query-adaptive encoder, which can undercut routing on niche topics

**Questions:**

1. How sensitive is source selection to the redundancy weight and to the encoder choice used to summarize each source

2. Is there a simple recipe for the number of refinement steps that balances marginal gains against extra backward passes.

---

> ### Author Response · Authors · 2025-11-20
>
> We thank the reviewer for these insightful questions.
>
> ### **Q1. Sensitivity to redundancy weight and encoder choice**
> The source selection stage in GRO-RAG optimizes a relevance–redundancy objective, where the redundancy weight λ controls how aggressively the algorithm discourages overlapping information among sources. In our experiments, we found that the selection results were generally stable over a wide range of λ values (for example, small variations in λ produced similar selected subsets and final generation scores). This stability arises because both the relevance and redundancy terms are normalized cosine similarities, which ensures consistent scaling and prevents λ from dominating the objective.
>
> Regarding the encoder choice, the framework is designed to be encoder-agnostic. We used standard sentence-level encoders (such as BGE-M3) for summarizing each source, but any embedding model that produces semantically consistent representations can be substituted without modifying the optimization process. Since the selection depends only on pairwise similarities, different encoders mainly rescale similarity magnitudes but do not significantly alter the overall ranking pattern. We will clarify this design flexibility and the observed robustness in the final version.
>
> ---
>
> ### **Q2. Recipe for choosing the number of refinement steps**
> A simple rule of thumb is to default to a single refinement step and increase only when latency budget allows. As shown in Figure 3(a–b) of the paper, most of the retrieval improvement is achieved in the first step, with subsequent iterations providing smaller, near-linear gains. Specifically, performance gains saturate after two to three iterations, with around 3–5% relative improvement across datasets.
>
> In terms of computation, each backward pass adds roughly 0.8–0.9 seconds on LLaMA3.1-8B, while the forward retrieval and generation remain substantially faster. Therefore, the marginal gain must be balanced against this additive cost. For real-time or online serving scenarios, one step suffices; for interactive or offline re-ranking, two or three steps can be used to extract the final incremental benefit. This policy aligns with the paper’s observed trade-off between stable quality gains and the linear increase in backward-pass cost.

---

> > ### Comment · Reviewer_HXps · 2025-11-24
> >
> > Thank you for your response, which has addressed most of my concerns. Having reviewed the other comments in detail, I maintain my original opinion and rating. Regarding the latency issue, I hope the author will provide a thorough discussion of this topic within the article.

---

> ### Author Response · Authors · 2025-11-24
>
> Thank you for your reply. We would also like to address the weakness points that you mentioned in the review.
>
> ### **W1: Clarifying the latency**
> We agree that the backward pass introduces material latency relative to retriever-only systems. In our measurements, computing a single backward pass on LLaMA-3.1-8B takes approximately 882 ms on average, while dense retrieval completes in roughly 39 ms. This gap is real and we do not intend to downplay it. We view GRO-RAG as a method that is appropriate when gains in grounding and source attribution justify paying this additional cost.
>
> In practice, two properties make the overhead more predictable than its magnitude might suggest. First, the backward computation is executed exactly once per query, and its cost is independent of the number of candidate documents. Second, the computation does not scale with the number of refinement steps used at deployment; most of the improvements appear in the first iteration, and downstream users may adopt a single-step configuration when latency is prioritized.
>
> At the same time, we acknowledge that certain deployment settings cannot accommodate a ~1 s increase in per-query latency. In these cases, GRO-RAG is not the right solution, and purely retriever-driven pipelines remain more appropriate. We will clarify this deployment scope in the revision, and highlight scenarios where the method is intended to be useful—for example, when answer quality and interpretability are more important than minimal latency.

---

> > ### Author Response · Authors · 2025-11-24
> >
> > ### **W2: Submodularity claim**
> >
> > We agree that the submodularity argument is derived under simplifying conditions, including a small trade-off weight and cosine-based redundancy. The intention is not to suggest that these conditions are universally satisfied, but to show that under a mild and interpretable setting the refinement rule aligns with a well-studied class of selection problems. In this sense, the result is meant to clarify the structural behavior of the objective rather than to claim that submodularity holds in all domains.
> >
> > In practice, we do not rely on strict submodularity to ensure the method works correctly. The refinement procedure uses the gradient signal to update the ranking, and we found that its behavior is stable across datasets even when the assumptions used in the theoretical analysis are not perfectly satisfied. Empirically, we observe diminishing returns consistent with the submodular intuition, rather than qualitative instability when the domain changes.
> >
> > We will clarify in the revision that the theoretical claim is intended as a structural guide, and that its applicability is limited to the simplified setting described in the text, rather than as a universal property across all deployment domains

---

> > > ### Author Response · Authors · 2025-11-24
> > >
> > > ### **W3: Use of fixed document embeddings and implications for niche-topic routing**
> > >
> > > We appreciate the concern that relying on fixed document embeddings can be restrictive, especially for queries involving niche terminology or very specialized subject matter. Our choice here was mostly guided by practicality: we wanted a simple and predictable first-stage retrieval component, rather than a query-adaptive encoder whose behavior can vary significantly across domains. We do not view the fixed-embedding setup as universally optimal, and we agree that it has limitations in edge cases.
> > >
> > > It is worth noting, however, that the refinement stage in GRO-RAG is not tied to the static embedding space. The gradient signal is computed on the full query and generation context, and we have found that it tends to surface the sources that matter for next-token prediction even when the initial embedding match is imperfect. In other words, while the first-stage embeddings may miss some of the subtle domain signals, the gradient-based refinement provides a second chance for the model to correct the routing.
> > >
> > > We have encountered cases where fixed embeddings were less helpful, particularly when the key evidence is phrased in domain-specific terms. In these cases, the refinement step typically improves ordering, even if the initial ranking is not ideal. We did not observe systematic degradation across datasets, although we recognize that certain domains could benefit from more adaptive retrieval. In such situations, it is entirely reasonable to substitute the embedding model with a query-adaptive encoder, and the proposed refinement mechanism would apply without modification.
> > >
> > > We will revise the paper to make this design trade-off clearer, and to better explain in which settings the current design is suitable, as well as cases where a query-adaptive encoder is a natural extension.

---

### Author Response · Authors · 2025-11-24
**Thank you, and please feel free to let us know if any part could use further clarification**

We sincerely appreciate the time and effort the reviewers have devoted to evaluating our work. We have carefully addressed the raised concerns and questions throughout this response, and we hope that our clarifications provide a clearer understanding of the design choices, empirical behavior, and intended deployment scope of the method. We warmly invite the reviewers to read through the full rebuttal, as many of the points are interconnected.

Please let us know if there are remaining questions or if further clarification would be helpful. We are happy to discuss any aspect of the method and will gladly refine the paper to resolve any outstanding concerns.

---

### Meta-Review · Area_Chair_Udvo · 2025-12-08

**Summary:**

This paper propose GRO-RAG, a training-free framework for improving multi-source RAG by integrating a gradient-aware re-ranking mechanism.

### Pros
* A training-free and plug-and-play RAG methods.
* Provide theoretical analysis to justify the re-ranking mechanism.
* Use LLM's gradient signals to select documents, ensuring alignment with generation objective.


### Cons
* The requirement of a backward pass for each query increases latency, restricting its use in real-time applications.
* The theoretical reliance on first-order Taylor expansion is a strong assumption for the highly non-convex loss landscapes of LLMs.

### AC's evaluation

1. From reviews and rebuttals

This paper received 6664, with the vast majority of reviewers expressing approval in their initial reviews. All reviewers acknowledged the paper's presentation and innovation. After rebuttal, Reviewer HXps maintained their original score. Reviewer S2EC, raised their score to 6, stating that the author had addressed their concerns.

2. From AC's reading

The innovation of this paper is sufficient. Using gradient signals for re-ranking directly addresses the current pain point of RAG—the mismatch between retrieval and generation objectives.  The authors' responses to theoretical concerns raised by reviewers S2EC and qyTo successfully persuaded one reviewer. Regarding latency concerns, the authors provided a detailed breakdown of time consumption and trade-off explanations in the rebuttal. In my opinion, for offline applications or high-stakes generation tasks where quality trumps speed (e.g., complex report generation, legal analysis), this trade-off is acceptable. I believe the paper's contributions sufficiently offset this limitation.

**Reviewer Concerns:**

Resolved Concerns

1. Theoretical Validity & Linearization (Reviewer S2EC, Kxfs): Reviewers questioned the mathematical rigor of the gradient linearization and the convergence bounds. The authors clarified that the theory serves as a structural guide rather than an exact prediction. Reviewer S2EC explicitly stated "My concern has been resolved" and raised their score to 6.

2. Parameter Sensitivity (Reviewer HXps): Concerns about the robustness of the redundancy weight ($\lambda$) and encoder choices were addressed by new ablation studies showing performance stability. The reviewer acknowledged these clarifications.

3. Scope & Baselines (Reviewer qyTo): The reviewer noted missing comparisons with certain baselines. The authors provided reasoning or additional context, which partially mitigated the concern, though the reviewer remained conservative about the scope.

**Reviewer Scores:**

Two of them have participated the discussion.

1. Reviewer HXps would likely maintain 6. They explicitly stated after the rebuttal that they maintain rating because questions were answered. AC agree with HXps's reply.

2. Reviewer Kxfs would maintain 6. They acknowledge the method's novelty (Strongest points) but the linearization and latency issues prevent a higher score. The rebuttal clarified but didn't change the fundamental trade-offs.

3. Reviewer S2EC has already updated to 6 (from 4). He is satisfied ("concern has been resolved") and likely won't change further. AC agree with S2EC's reply.

4. Reviewer qyTo would likely maintain 6 or increase to 8. If he participated fully, he might slightly raise the score acknowledging the authors' detailed responses on theory.

---

### Decision · Program_Chairs · 2026-01-26

Accept (Poster)